# Chondrogenic Potential of Human Umbilical Cord Mesenchymal Stem Cells Cultured with Exosome-Depleted Fetal Bovine Serum in an Osteoarthritis Mouse Model

**DOI:** 10.3390/biomedicines10112773

**Published:** 2022-11-01

**Authors:** Yu-Hsun Chang, Kun-Chi Wu, Dah-Ching Ding

**Affiliations:** 1Department of Pediatrics, Hualien Tzu Chi Hospital, Buddhist Tzu Chi Medical Foundation, Tzu Chi University, Hualien 970, Taiwan; 2Department of Orthopedics, Hualien Tzu Chi Hospital, Buddhist Tzu Chi Medical Foundation, Tzu Chi University, Hualien 970, Taiwan; 3Department of Obstetrics and Gynecology, Hualien Tzu Chi Hospital, Buddhist Tzu Chi Medical Foundation, Tzu Chi University, Hualien 970, Taiwan; 4Institute of Medical Sciences, Tzu Chi University, Hualien 970, Taiwan

**Keywords:** mesenchymal stem cells, umbilical cord, osteoarthritis, cartilage, exosome, regeneration

## Abstract

Osteoarthritis (OA) is characterized by the loss of articular cartilage and is also an age-related disease. Recently, stem cell therapy for cartilage repair has emerged. The stem cells need to be cultured with a fetal bovine serum (FBS)-supplemented medium. The effect of FBS-containing exosomes on the differentiation of human umbilical cord mesenchymal stem cells (HUCMSCs) is unknown. The morphology, proliferation, surface marker expressions, and trilineage differentiation ability of two groups of HUCMSCs, cultured with conventional (FBS) and exosome-depleted FBS (Exo(-)FBS), were evaluated. In a mouse OA model after two groups of HUCMSCs transplantation, the rotarod activity, histology, and immunohistochemistry (type II collagen, aggrecan, IL-1β, and MMP13) of the cartilage were evaluated. The Exo(-)FBS-cultured HUCMSCs, like FBS-cultured HUCMSCs, displayed classic MSC characteristics, including spindle-shaped morphology, surface marker expression (positive for CD44, CD73, CD90, CD105, and HLA-ABC and negative for CD34, CD45, and HLA-DR), and trilineage differentiation (chondrogenesis, osteogenesis, and adipogenesis). The Exo(-)FBS-cultured HUCMSCs proliferated significantly slower than those of the FBS-cultured HUCMSCs (*p* < 0.01). The trilineage gene expression of PPAR-γ, FABP4, APAL, type II collagen, aggrecan, and SOX9 was significantly increased in the Exo(-)FBS-cultured HUCMSCs than in the FBS-cultured HUCMSCs and undifferentiated controls. The Exo(-)FBS- and FBS-cultured HUCMSCs-transplanted mice showed a better rotarod activity than in the control OA mice (*n* = 3 in each group). A significant histological improvement in hyaline cartilage destruction after the transplantation of both types of FBS-cultured HUCMSCs was noted when compared with the OA knees. The Exo(-)FBS-cultured HUCMSCs-transplanted knees showed a higher International Cartilage Repair Society histological score (*p* < 0.05), staining intensity of type II collagen (*p* < 0.01), and aggrecan (*p* < 0.01) than in the control knees. Moreover, both types of the FBS-cultured HUCMSCs-transplanted knees significantly decreased the expression of MMP13 and IL-1β compared to that in the OA knees (*p* < 0.01). The Exo(-)FBS-cultured HUCMSCs harbor chondrogenic potential and attenuated cartilage destruction in a mouse OA model. Our study provides a basis for future clinical trials using Exo(-)FBS-cultured stem cells to treat OA.

## 1. Introduction

Osteoarthritis (OA) is characterized by the loss of articular cartilage and is an age-related disease [1]. Patients with OA experience disability and joint pain. Joints that bear weight and undergo excessive and repetitive stress are predisposed to the development of OA. Cartilage does not have regenerative capacity and responds poorly to conventional therapies, such as rehabilitation and pharmacotherapy. Surgical procedures, such as arthroplasty or total joint replacement, are ultimately used to treat OA. Recently, stem cells developed from various tissues, such as adipose tissue, umbilical cord, and bone marrow, have been used to support cartilage repair [2,3,4]. Stem cells can differentiate into chondrocytes and integrate into the cartilage [2] and perform immunomodulation to decrease catabolic and inflammatory cytokine production in OA cartilage [5].

Human umbilical cord mesenchymal stem cells (HUCMSCs) are an emerging source of stem cells [6]. The process of obtaining HUCMSCs is a non-invasive procedure. HUCMSCs possess a faster self-renewal characteristic, can undergo trilineage differentiation, promote tissue repair, and harbor immunomodulatory ability [6]. Therefore, HUCMSCs are attractive cell sources for autologous or allogeneic transplantation in degenerative diseases, including neurodegenerative diseases and osteoarthritis [6].

Fetal bovine serum (FBS) was required to maintain cell growth. However, exosomes in conventional FBS influence the experimental results, including transplantation efficacy [7]. Exosome-depleted FBS (Exo(-)FBS) has reduced exosome content compared to conventional FBS. More than 90% of exosomes are depleted in the commercial Exo(-)FBS (ThermoFisher, Waltham, MA, USA). FBS-containing exosomes have been reported to affect the anchorage growth in breast cancer cells [8]. Moreover, in the A549 lung cancer epithelial cell line, FBS induces a migratory phenotype [9]. FBS-derived exosomes have the same effect on A549 cells [9]. In a previous study using lipopolysaccharide-induced immune response, the authors found that FBS exosomes reduce interleukin (IL)-6, TNF-α, and IL-1β levels [10]. Therefore, Exo(-)FBS has been developed for use in cell culture experiments [11]. However, Exo(-)FBS has a disadvantage as it affects the cell proliferation rate and differentiation capability [12].

The culture of HUCMSCs needs FBS to maintain proliferation and differentiation [13]. However, above all, the exosomes in FBS may affect the proliferation and differentiation capability of HUCMSCs. The HUCMSCs can secrete exosomes to affect many cell types in the microenvironment [14,15]. We expected the benefit of Exo(-)FBS on the differentiation of HUCMSCs, including no influence on HUCMSCs’ exosome secretion and maximizing self-secreted exosomes’ functionality, which may enhance the differentiation capability of HUCMSCs.

Above all, animal exosomes may influence cell proliferation and differentiation. Exosomes derived from mesenchymal stem cells (MSCs) or chondrocytes may promote chondrogenesis or chondrocyte proliferation [14,16]. However, the influence of animal exosomes in FBS on the chondrogenesis of HUCMSCs is unknown. We hypothesized Exo(-)FBS might enhance the chondrogenesis of HUCMSCs due to no influence from animal exosomes.

This study aimed to examine the feasibility of Exo(-)FBS-cultured HUCMSCs for chondrocyte differentiation and transplantation into an OA mouse model. Native FBS-cultured HUCMSCs were used as the control.

## 2. Materials and Methods

### 2.1. Ethics

The human sample collection and all experiments, procedures, and protocols were approved by the Research Ethics Committee of Hualien Tzu Chi Hospital (IRB 110-195-C). All study participants gave their written informed consent.

### 2.2. Derivation and Culture of Human Umbilical Mesenchymal Stem Cells

The HUCMSCs’ derivation and culture protocol has been reported previously [17]. Briefly, after childbirth, a 20-cm length of the human umbilical cord was collected in sterile boxes containing 20 mL of normal saline (Taiwan BioTech, Taoyuan, Taiwan). After isolating the vessels and amniotic membranes, the remaining Wharton’s jelly was cut into small pieces using scissors and treated with type I collagenase (Sigma, St Louis, MO, USA), and incubated for 14–18 h in 5% CO_2_ at 37 °C. The tissue fragments were then cultured in Dulbecco’s modified Eagle medium-low glucose (DMEM-LG) (Gibco, Grand Island, NY, USA) supplemented with 10% FBS (Biological Ind., Kibbutz, Israel) and antibiotics. The tissue fragments were then left undisturbed for 5–7 days to allow for the cells’ migration from the tissue fragments. The resulting cells were designated as passage 1 (P1).

In this experiment, HUCMSCs were used for the experiments between P3 and P8. The below experiments were performed by culturing HUCMSCs with native FBS and Exo(-)FBS.

### 2.3. Exosome-Depleted FBS (Exo(-)FBS)

Exosome-depleted FBS was purchased from Gibco (A2720801, ThermoFisher Scientific, Waltham, MA, USA). The depleted Exo(-)FBS has the most endogenous exosomes (>90%) and is ready for use in cell culture exosome isolation assays. The Exo(-)FBS is ready for use and eliminates the risk associated with the preparation of Exo(-)FBS by ultracentrifugation.

### 2.4. XTT Assay

An XTT kit was used for the cell proliferation assay. Different FBS-cultured HUCMSCs were seeded at a density of 2 × 10^3^ cells/well in a 96-well plate. The final volume was 100 μL of culture medium in one well. To proceed with the proliferation assay, the cells were incubated with 150 μL XTT solution (Biological Industries, Kibbutz Beit Haemek, Israel) at 37 °C for 3 h. A microplate reader (Model 3550, Bio-Rad, Hercules, CA, USA) was used for the absorbance reading (450 nm). Proliferation curves were expressed as optical density values and were constructed on days 0, 3, and 7.

### 2.5. Flow Cytometry

Flow cytometry was used to evaluate the surface marker expression of HUCMSCs (P3-4) cultured with different types of FBS. HUCMSCs were separated by adding phosphate-buffered saline (PBS) containing Accutase (Millipore, Billerica, MA, USA). Then, cells were washed with 2% bovine serum albumin and 0.1% sodium azide (Sigma-Aldrich, St. Louis, MO, USA) containing PBS. The cells were then incubated with primary antibodies conjugated with phycoerythrin or fluorescein isothiocyanate including HLA-ABC, HLA-DR, CD34, CD44, CD73, CD90, and CD105 (BD Biosciences, Franklin Lakes, NJ, USA). A Becton Dickinson flow cytometer (Becton Dickinson, San Jose, CA, USA) was used for analysis.

### 2.6. Annexin V/PI Staining

The Annexin V/PI (propidine iodide) Apoptosis Detection kit (Invitrogen) was used to detect the cells’ apoptosis. Briefly, the cells were plated into 6-well plates. After 24 h of culture in Exo(-)FBS or FBS, the cells were trypsinized, cold PBS washed twice, and resuspended in binding buffer (195 µL). Another 5 µL Annexin V-FITC and 10 µL PI working solution were added and then incubated for 15 min in the dark at room temperature. The fluorescence of the cells was shown by a fluorescence microscope (Zeiss Axio Imager Z1, Carl Zeiss, Oberkochen, Germany).

### 2.7. Induction to Adipocytes

The adipogenic medium was composed of DMEM supplemented with 10% exosome +/− FBS, 0.5 mmol/L isobutylmethylxanthine, 5 µg/mL insulin, 60 μmol/L indomethacin, and 1 µmol/L dexamethasone (Sigma, St. Louis, MO, USA). A total of 5 × 10^4^ HUCMSCs were plated in one well of a 12-well plate and fed adipogenic medium for 14 days. The medium was replaced every 3 days. Oil red O (Sigma, St. Louis, MO, USA) was used for staining the adipocytes. The staining cell pictures were taken by a microscope (Nikon, Tokyo, Japan).

### 2.8. Induction to Osteoblast

The osteogenic medium was composed of DMEM supplemented with 10% exosome +/− FBS, 10 mmol/L β-glycerol phosphates, 0.1 µmol/L dexamethasones, and 50 μmol/L ascorbic acids. The HUCMSCs (1 × 10^4^) were plated in one well of a 12-well plate with the above medium for 14 days and the medium was replaced every 3 days. Alizarin red (Sigma, St. Louis, MO, USA) was used for staining the osteoblast. The cell picture was taken by a microscope (Nikon, Tokyo, Japan).

### 2.9. Induction to Chondrocyte

The chondrogenic medium was composed of DMEM supplemented with exosome +/− FBS, 6.25 μg/mL insulin, 10 ng/mL transforming growth factor-β1, and 50 μg/mL ascorbic acid-2-phosphate (Sigma, St. Louis, MO, USA). For chondrogenesis, the pellet culture method was used. A total of 1 × 10^6^ HUCMSCs were plated in a 15-mL conic tube (BD) filled with 2 mL chondrogenic medium for 21 days and the medium was replaced every 2 days. The resulting pellet was pictured and then fixed in 4% paraformaldehyde at 4 °C for 24 h. After that, PBS was used for washing the pellets and transferred to 70% ethanol. Histology (hematoxylin and eosin (H & E) staining, safranin O staining) was applied to characterize the differentiated chondrocytes.

### 2.10. Quantitative RT-PCR Analyses

After the trilineage differentiation of the HUCMSCs, the RNeasy Protect Mini Kit with on-column RNase-free DNase treatment (Qiagen, Hilden, Germany) was used to extract total RNA from cells. An amount of 30 mL RNase-free water was used for eluted RNA. An amount of 8 mL eluate was used for reverse transcription using the SuperScript III One-Step RT-PCR kit (Invitrogen, Grand Island, NY, USA) to synthesize cDNA. FastStart SYBR Green QPCR Master (ROX, Roche, Indianapolis, IN, USA) on a quantitative real-time PCR detection system (ABI Step One Plus system, Applied Biosystems, Foster City, CA, USA) was used to amplify real-time PCR, using 2mL of the cDNA product. The primer sequences were as follows (all at 150 nM final concentration):

For adipogenesis were *PPAR-γ* (forward, 5′-AGCCTCATGAAGAGCCTTCCA-3′; reverse, 5′-TCCGGAAGAAACCCTTGCA-3′) and *FABP4* (forward, 5′-ATGGGATGGAAAATCAACCA-3′; reverse, 5′-GTGGAAGTGACGCCTTTCAT-3′); for osteogenesis were *ALPL* (forward, 5′-CCACGTCTTCACATTTGGTG-3′; reverse, 5′-GCAGTGAAGGGCTTCTTGTC-3′) and *RUNX2* (forward, 5′-CGGAATGCCTCTGCTGTTAT-3′; reverse, 5′-TTCCCGAGGTCCATCTACTG-3′), for chondrogenesis were *SOX9* (chondrogenic marker) (forward, 5′-ACACACAGCTCACTCGACCTTG-3′; reverse, 5′-GGGAAT TCTGGTTGGTCCTCT-3′); type II collagen (*COL2A1*) (chondrogenic marker) (forward, 5′-GGACTTTTCTCCCCTCT CT-3′; reverse, 5′-GACCCGAAGGTCTTACAGGA-3′), and aggrecan (*ACAN*) (chondrogenic marker) (forward, 5′-GAGATGGAGGGTGAGGTC-3′; reverse 5′-ACGCTGCCTCGGGCTTC-3′). The glyceraldehyde-3-phosphate dehydrogenase (*GAPDH*) (housekeeping gene and internal control) (forward, 5′-GAAGGTGAAGGTCGGAGTC-3′; reverse, 5′-GAAGATGGT GATGGGATTTC-3′) was used as an internal control. The 2^-ΔΔCt^ method was used for quantification of the threshold cycle (Ct) value for each gene compared with GAPDH [18].

### 2.11. Type VII Collagenase-Induced Osteoarthritis in a Mouse Model

The Institutional Animal Care and Use Committee of Hualien Tzu Chi Hospital approved all the experiments on the animals.

Immune cells play an important role in the OA microenvironment, so immunocompetent mice were chosen for the experiments. Nine female B6 mice aged 6–8 weeks and weighing 18–22 g were divided into three groups: the control group injected with normal saline (*n* = 3), the FBS-cultured HUCMSCs-treated group (*n* = 3), and the Exo(-)FBS-cultured HUCMSCs-treated group (*n* = 3). The OA cartilage condition was created by the intra-articular injection of type VII collagenase (C0773, Sigma-Aldrich, St Louis, MO, USA) into both hind legs. A 30-gauge needle (BD Pharmingen) was administered with 8 μL of collagenase dissolved in normal saline at a dose of 12 units per joint.

Seven days after lesioning, three groups of mice were injected with normal saline or exosome +/− FBS-cultured HUCMSCs. In the control group of mice, both knees were injected with 50 μL of normal saline.

### 2.12. Transplantation of Human Umbilical Cord Mesenchymal Stem Cells

Seven days after the collagenase injection, the mice were anesthetized with ketamine (50 mg/kg) and xylazine (15mg/kg) injection intraperitoneally. A total of 4 × 10^5^ exosome+/− FBS-cultured HUCMSCs in 50 μL PBS were injected intra-articularly in the hind legs. After transplantation, the mice were permitted to move and eat freely after recovery from anesthesia.

### 2.13. The Assessment of the Rotarod Behavior

The purpose of the rotarod test was to measure the stay time on the rotating rod of the mice that received stem cell treatment or not. The stay time may be related to the recovery condition of the knee injury after treatment.

The rotarod (3376-4R, TSE Systems, Chesterfield, MO, USA) training began 3 days before the collagenase injection. After a steady running experience for each mouse, the mice received the collagenase injection. After lesioning, the mice were tested for rotarod running at the light phase at 7-day intervals. The mice were permitted to become familiar with the testing environment for at least 30 min. The mice were positioned on a wheel of a rotarod and forced ambulation at a speed of 20 rpm for a maximum of 120 s per test. The test was repeated 5 times during the test time period (days 0, 7, 14, and 28). The data of the last three time periods were compared to the data of day 0. The duration of rotarod running was presented as the meantime on the rotating bar over five test trials.

### 2.14. Tissue Collection

The mice were euthanized after the experiments finished on day 28. The surface of the joint cartilage was inspected closely. The proximal tibial and distal femoral plateaus were harvested and fixed in the 10% buffered formalin (Sigma, St. Louis, MO, USA) for 48 h. Then, the specimen was decalcified with 10% ethylenediaminetetraacetic acid (Gibco, Waltham, MA, USA) for 14 days. After the cartilage become softened, the specimen was cut into 4 pieces. The specimen was embedded in paraffin, and serial sagittal sections were done. The sections were stained by H & E staining (Sigma, St. Louis, MO, USA) and Safranin O (Sigma, St. Louis, MO, USA). The microscopic pictures were taken under the microscope (Nikon, Tokyo, Japan).

### 2.15. Histological Evaluation

The International Cartilage Repair Society (ICRS) scoring system (including surface, matrix, cell distribution, cell population viability, subchondral bone, and cartilage mineralization) was used to evaluate the histologic change of the cartilage [19]. A total score of 0–18 was assigned to the above 6 categories. The better the function of the cartilage was indicated by a higher score. Two researchers independently evaluated the scores and mean scores were calculated.

### 2.16. Immunohistochemistry

Immunohistochemistry (IHC) was used to assess the expression of the cartilage markers (type II collagen and aggrecan), a catabolic marker (matrix metallopeptidase (MMP)13), and an inflammatory marker (IL-1β). All primary antibodies were purchased from GeneTex with a 1:100 dilution. The horseradish peroxidase-linked secondary antibody was applied and incubated with a substrate of diaminobenzidine tetrahydrochloride (Abcam, Cambridge, UK). The positive staining area presented brown in color. The sections were pictured by a light microscope (Nikon, Tokyo, Japan). The intensities of the above four proteins were quantified by using ImageJ software (NIH, Bethesda, MD, USA) [20].

### 2.17. Statistical Analysis

SPSS software (version 24, IBM, New York, NY, USA) was used for statistical analysis. The results were represented as mean ± standard error of the mean.

Raw data from qRT-PCR, Rotarod duration, histological scores, and quantification of IHC were analyzed using the ANOVA test with the Bonferroni post hoc test. The difference between the two groups was measured by Student’s *t*-test. *p* < 0.05 was considered to reach a statistical significance.

## 3. Results

### 3.1. HUCMSCs Showed Characteristics of MSCs

#### 3.1.1. Morphology, Proliferation, and Surface Marker Expression Levels

The HUCMSCs cultured with FBS or Exo(-)FBS showed the same morphology (fibroblastic and spindle shape) on days 1 and 4 (Figure 1A). The cell proliferation curve of HUCMSCs cultured with Exo(-)FBS was significantly slower than with FBS (*p* < 0.01) (Figure 1B). The flow cytometry of HUCMSCs cultured with Exo(-)FBS and FBS showed the same expression pattern. They were negative for CD34, CD45, and HLA-DR and positive for CD44, CD73, CD90, CD105, and HLA-ABC (Figure 1C). The CD73 expression was lower in Exo(-)FBS than in FBS (35% vs. 97.2%). The dot-plot figure showed differences in the size and granularity of the HUCMSCs cultured with Exo(-)FBS and FBS (Figure 1D). No difference in cell viability in both the Exo(-)FBS- and FBS-cultured HUCMSCs was demonstrated by Annexin V/PI double staining (Figure 1E).

#### 3.1.2. Adipocyte and Osteoblast Differentiation of HUCMSCs

After adipogenic differentiation for 14 days, exosome +/− FBS-cultured HUCMSC-differentiated adipocytes were positive stained with oil red O (Figure 2A). Figure 2B shows the expression of adipogenesis-related genes, as assessed by qRT-PCR. PPARγ and FABP4 expression levels were significantly higher in Exo(-)FBS-cultured HUCMSC-differentiated adipocytes than in FBS (*p* < 0.001). Osteogenesis was evaluated after 14 days of differentiation and stained with alizarin red. FBS- and Exo(-)FBS-cultured HUCMSC-differentiated osteoblasts were alizarin red staining positive and showed osteoblast-related gene expressions (Figure 2C,D). APAL expression was significantly higher in Exo(-)FBS-cultured HUCMSC-differentiated osteoblasts than in FBS-cultured cells (Figure 2D). The absorbance of oil red and alizarin red is illustrated in Figure 2E,F. The amounts of oil red and alizarin red in Exo(-)FBS-HUCMSCs were less than FBS-HUCMSCs, indicating fewer survival cells expressing two stainings.

#### 3.1.3. Chondrogenesis of FBS- and Exo(-)FBS-Cultured HUCMSCs

HUCMSCs were cultured with FBS or Exo(-)FBS and subjected to chondrogenic differentiation in vitro for 21 days. The Exo(-)FBS-cultured HUCMSC-differentiated chondrocytes had a significantly increased expression of chondrocyte markers (type II collagen and aggrecan) but not SOX9 compared to the FBS-cultured cells (Figure 3A). Figure 3B–D shows gross images, histology, and safranin O staining of the pellet from exosome +/− FBS-cultured HUCMSCs. Owing to the mRNA level correlated with cell viability [21], we used the mRNA level to indicate cell viability after chondrogenesis. Figure 3E shows the concentration of mRNA from two kinds of FBS-cultured HUCMSCs after chondrogenesis, which might indicate the cell number of Exo(-)FBS-HUCMSCs was less than FBS-HUCMSCs after chondrogenesis (Figure 3E). Taken together, these results indicate that Exo(-)FBS-cultured HUCMSCs underwent chondrogenesis after chondrogenic differentiation.

### 3.2. Rotarod Behavior after Treatment

To test the mice’s walking ability after a knee injury and stem cell treatment, the rotarod test was used to measure the mice’s ability to maintain themselves upright on a rotating rod. Stay time on a rotating rod was recorded.

On day 7 after the collagenase digestion of the knee cartilage, there was a significantly decreased in-running duration in all three groups compared to day 0 (Figure 4). The duration of the control group was significantly decreased on days 7 (45%), 14 (30%), 21 (20%), and 28 (15%) when compared with day 0. In contrast, the duration in the FBS-HUCMSCs group was significantly improved to 90% on day 28 compared to day 0 and was better than that in the OA control group (*p* < 0.01). The duration in the Exo(-)FBS-HUCMSCs group was significantly improved to 70% on day 28 compared to day 0, and better than that in the OA control group (*p* < 0.05) (Figure 4). Taken together, both the FBS- and Exo(-)FBS-cultured HUCMSCs-treated mice showed an improvement in the rotarod duration than in the control mice.

### 3.3. Histological Evidence of Cartilage Repair with HUCMSCs in Collagenase-Treated Mice

The histology and IHC were used for evaluating the effect of HUCMSCs transplantation on cartilage destruction in collagenase-treated mice. A greater cell loss in the knee cartilage in the control group compared with the HUCMSCs (cultured with FBS and Exo(-)FBS) transplantation group was shown by H & E staining (Figure 5A). Glycosaminoglycans (GAGs) content in the cartilage was also reduced in the control group compared to the HUCMSCs transplantation group (cultured with FBS and Exo(-)FBS) and was shown by Safranin O staining (Figure 5A). The ICRS grading system was used to quantify histological changes in the cartilage [22]. Higher scores were observed in the Exo(-)FBS-cultured HUCMSCs transplantation group than in the OA control group (*p* < 0.05). In summary, histological evidence indicates that mice transplanted with both FBS- and Exo(-)FBS-cultured HUCMSCs showed reduced cell and GAG loss in collagenase-destructed cartilage.

### 3.4. Increased Expression of Type II Collagen and Aggrecan after Transplantation of HUCMSCs

In the immunohistochemical analysis of mouse knees for aggrecan (Figure 6A) and type II collagen (Figure 6C), the destructed cartilage in the control mice was nearly pale, indicating the absence of hyaline cartilage. In turn, the knee transplanted with HUCMSCs showed a greater distribution of staining, indicating hyaline cartilage (Figure 6A,C). The quantification of the staining intensity of aggrecan (*n* = 3 in each group, Figure 6B) was significantly higher in the FBS-cultured HUCMSC-treated group than in the other two groups (*p* < 0.01). The quantification of type II collagen was significantly higher in both FBS- and Exo(-)FBS-cultured HUCMSC-treated groups (*n* = 3 in each group) than in the control group (*p* < 0.01 and 0.001) (Figure 6D).

### 3.5. Decreased Expression of MMP13 and IL-1β in HUCMSC-Treated Cartilages

To test whether the catabolic effect and inflammation were decreased in HUCMSC-treated cartilage, IHC with MMP13 and IL-1β was performed. MMP13 expression was significantly decreased in the cartilage after FBS- and Exo(-)FBS-cultured HUCMSC transplantation (*p* < 0.01) (Figure 7A,B, *n* = 3). IL-1β was also significantly decreased in the cartilage after FBS- and Exo(-)FBS-cultured HUCMSC transplantation (*p* < 0.01) (Figure 7C,D).

## 4. Discussion

An in vitro study showed that HUCMSC cultured with Exo(-)FBS had a slower proliferation rate than those cultured with FBS. The trilineage differentiation ability was the same for FBS- and Exo(-)FBS-cultured HUCMSCs. The typical OA findings with cartilage destruction were presented in a mouse model treated with collagenase. HUCMSCs cultured with FBS- or Exo(-) FBS-treated mice showed significant improvement in rotarod movements and histological findings in the joints. There was a significantly increased expression of aggrecan and type II collagen, and a decreased expression of an inflammatory marker (IL-1β) and a catabolic marker (MMP13) in both the FBS- or Exo(-)FBS-cultured HUCMSC-transplanted cartilage.

Exo(-)FBS may alter cell proliferation [23]. Eitan et al. reported that Exo(-)FBS reduces cell proliferation and viability (HeLa, HEK-293T, N2a mouse neuroblastoma cells, and SY5Y human neuroblastoma) [24]. Exosomes derived from FBS promote the anchorage-independent growth of breast cancer cells [8]. Exo(-)FBS also hampered cell proliferation in the primary cell culture system [23]. The affected primary culture cells include primary human myoblast [12], cardiac progenitor cells [7], and primary mouse astrocytes [25]. Suboptimal growth and viability of primary mouse astrocytes cultured by Exo(-)FBS than native FBS were also demonstrated [25]. The cause of decreased cell proliferation and viability may be due to cultures with Exo(-)FBS [23]. Aswad et al. found downregulated cell proliferation-related genes (e.g., SIRT1 and CCND1) in myoblast cultured with Exo(-)FBS [12]. Moreover, exosomes derived from FBS carried molecules associated with cell survival (e.g., sonic hedgehog, catalase, survivin, Wnt, TGF-beta, heat shock protein, and superoxide dismutase) [24,26]. Our results were concordant with the previous studies that the proliferation of HUCMSCs is impeded by Exo(-)FBS compared to FBS.

Besides impaired cell growth, other cell characteristics, including differentiation, migration, and inflammation, may also be impeded by Exo(-)FBS-cultured condition. The previous study showed that myoblasts could not differentiate into myotubes after culturing with Exo(-)FBS [12]. They concluded that exosome-depleted serum altered the muscle cells’ phenotype [12]. Angelini et al. reported that using Exo(-)FBS-cultured cardiac progenitor cells impedes cardiosphere formation [7]. Shelke et al. reported that airway epithelial cell migration was inhibited by culturing with Exo(-)FBS [9]. Our study agreed with the previous studies that Exo(-)FBS might promote chondrogenesis under chondrogenic differentiation. Taken together, Exo(-)FBS may affect the differentiation ability of stem cells.

Exosomes from FBS also affected immune cell functions and inflammatory response [10,23]. Exo(-)FBS-cultured primary microphages increase proinflammatory cytokines after lipopolysaccharide stimulation [10]. Exo(-)FBS-cultured human immunodeficiency virus (HIV) infected T lymphocytes, increased infectivity, cell aggregation, and the release of HIV-1 [27]. In our study, we found IL-1β decreased expression on damaged cartilage after Exo(-)FBS-cultured HUCMSCs transplantation in a mouse OA model.

Exosomes may affect chondrogenesis. Kim et al. reported that chondrocytes- and MSCs-derived exosomes affect cell proliferation and chondrogenesis [28]. Wu et al. reported that infrapatellar fat pad MSCs-derived exosomes protect against cartilage destruction by inhibiting the mTOR signaling pathway [29]. Shen et al. reported that hypoxia-conditioned MSC-derived exosomes promote chondrogenesis in hydrogels [30]. Although we did not evaluate the effects of exosomes derived from HUCMSCs cultured with Exo(-)FBS, the therapeutic effect may partially depend on HUCMSC-derived exosomes.

Our study used type II collagen and aggrecan as the marker of chondrogenesis. Type II collagen is the main component of the cartilage matrix [31]. Type II collagen can form complex scaffolds extracellularly to provide chondrocytes, extracellular matrix, and anchoring growth factors [32]. Aggrecan, a large proteoglycan, can bear keratan sulfate and chondroitin sulfate chains that compose articular cartilage, making it able to bear loads of compression [33]. Aggrecan interacts with hyaluronan to form an extracellular matrix and proteoglycan aggregates [33]. Therefore, type II collagen and aggrecan can be used as markers of chondrogenesis.

IL-1β can be produced by many types of cells and plays an important role in OA pathogenesis [34]. The mechanism of IL-1β on chondrocyte disruption is by increasing MMPs and aggrecanase and decreasing chondrogenic extracellular matrix synthesis [34]. Immune cells, including macrophages, neutrophils, and lymphocytes, can be attracted to OA sites and secrete IL-1β to aggravate OA pathologies [34]. MMP13 is the primary MMP to cleave type II collagen involving cartilage degradation [35]. Our results showed the decreasing IL-1β and MMP13 expression of the destructed cartilage after the transplantation of Exo(-)FBS-cultured HUCMSCs in a mouse OA model.

As mentioned above, FBS may influence MSC proliferation and differentiation. Therefore, lowering the influence of FBS can be achieved by using Exo(-)FBS. Although clinical-grade FBS can be used in clinical trials, exosomes in FBS might interfere with the results. We proved that Exo(-)FBS might enhance chondrogenesis or cartilage repair.

The strength of our study is the exploration of the chondrogenic effect of HUCMSCs cultured with Exo(-)FBS. To our knowledge, the chondrogenic phenotype of MSCs in Exo(-)FBS conditions has never been reported.

## 5. Conclusions

We demonstrate that the chondrogenesis of Exo(-)FBS-cultured HUCMSCs might be achieved. Exo(-)FBS-cultured HUCMSCs exhibit chondrogenesis and protection capability in in vitro and in vivo models. Our results might provide a foundation for clinical trials of stem cells cultured in Exo(-)FBS.

## Figures and Tables

**Figure 1 biomedicines-10-02773-f001:**
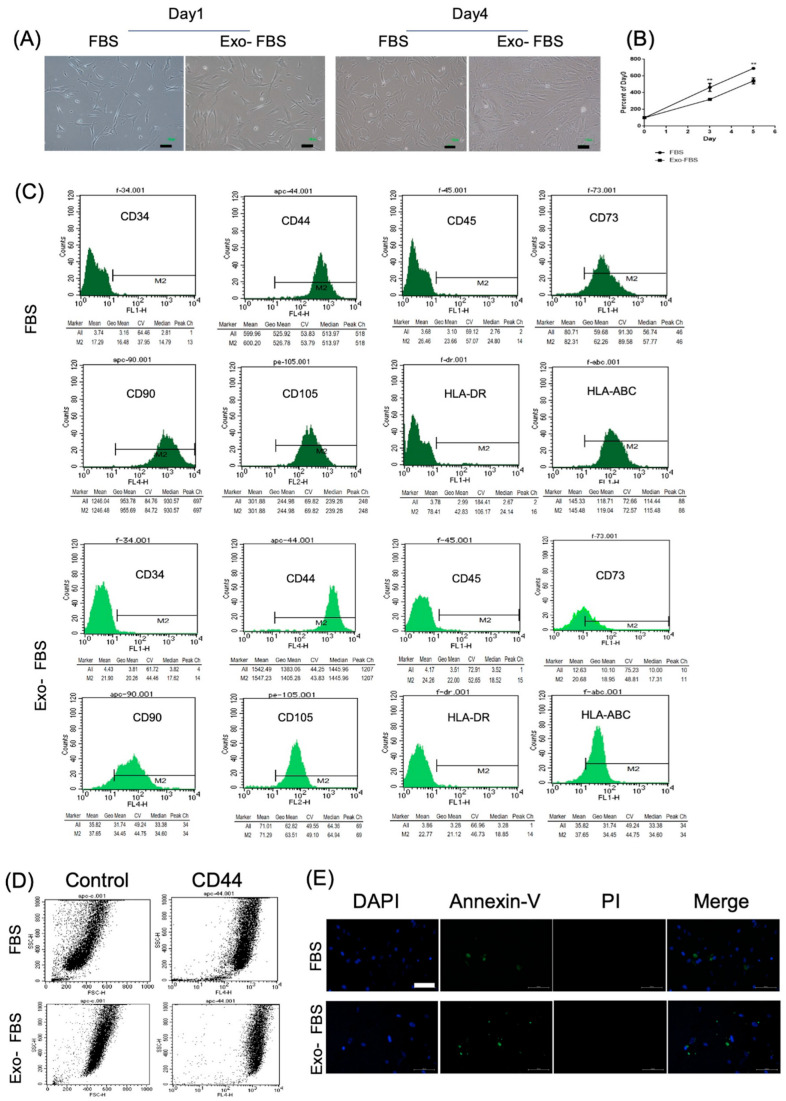
Morphology, proliferation, and surface markers of HUCMSCs cultured with exosome-depleted (Exo(-)FBS) and conventional fetal bovine serum (FBS). (**A**) Morphology of the Exo(-)FBS- and FBS-cultured HUCMSCs showed fibroblastic and spindle-shaped on days 1 and 4. Scale bar = 100 μm. (**B**) Cell proliferation curve of Exo(-)FBS- and FBS-cultured HUCMSCs. ** *p* < 0.01. Representative results from 3 experiments. (**C**) Flow cytometry of Exo(-)FBS- and FBS-cultured HUCMSCs. They are negative for CD34, CD45, and HLA-DR and positive for CD44, CD73, CD90, CD105, and HLA-ABC and Exo-: exosome-depleted FBS. HUCMSCs were derived from the same donor. (**D**) Dot plot FSC/SSC (forward scatter (size)/side scatter (granularity)) of CD44 expression in HUCMSCs culture with FBS and Exo(-)FBS. (**E**) HUCMSCs cultured with FBS or Exo(-)FBS were stained with FITC-Annexin V/PI and analyzed by fluorescence microscopy. Scale bar = 100 μm. DAPI: nuclear staining, PI: propidium iodide.

**Figure 2 biomedicines-10-02773-f002:**
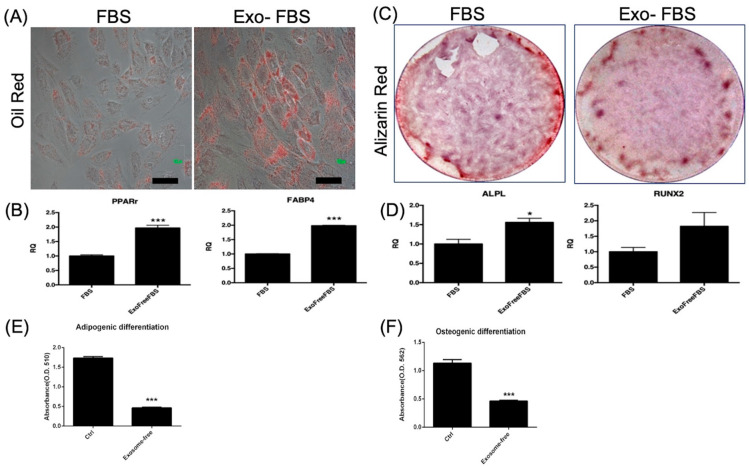
Adipogenesis and osteogenesis of HUCMSCs cultured with exosome-depleted and conventional fetal bovine serum (FBS). (**A**) Adipogenesis for 2 weeks and HUCMSCs-differentiated adipocytes showed positive staining of oil red. Scale bar = 100 μm. (**B**) Quantitative polymerase chain reaction (qPCR) analysis of adipocyte gene expressions (PPAR-γ, FABP4). *** *p* < 0.001. (**C**) Osteogenesis for 2 weeks and HUCMSCs-differentiated osteoblasts showed positive staining of alizarin red. Scale bar = 100 μm. (**D**) qPCR analysis for osteoblast gene expressions (APAL, RUNX2). * *p* < 0.05. (**E**) Absorbance (O.D. = 510 nm) of oil red. (**F**) Absorbance (O.D. = 510 nm) of alizarin red. *** *p* < 0.001. Representative results from 3 experiments. HUCMSCs were derived from the same donor.

**Figure 3 biomedicines-10-02773-f003:**
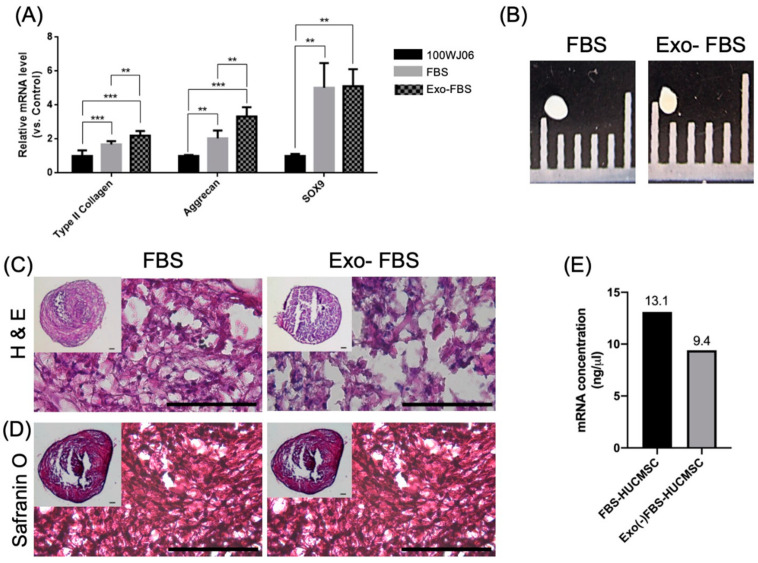
Chondrocyte gene expressions and immunohistochemistry (IHC) of exosome +/− FBS-cultured HUCMSCs. (**A**) qRT-PCR shows chondrogenic gene expressions (SOX9, aggrecan, and type II collagen) in exosome +/− FBS-cultured HUCMSCs cultured after chondrogenesis. ** *p* < 0.01, *** *p* < 0.001. Representative results from 3 experiments. (**B**) Gross picture of exosome +/− FBS-cultured HUCMSCs-differentiated cartilage pellet. Scale = 1 mm. (**C**) Hematoxylin and eosin staining of the pellet. (**D**) Safranin O staining of the pellet differentiation from HUCMSCs. Inlet images showed the original size. Scale bar = 100 μm. (**E**) mRNA concentration (ng/μL) after chondrogenic differentiation (*n* = 1). ** *p* < 0.01, and *** *p* < 0.001. HUCMSCs were derived from the same donor (101WJ06).

**Figure 4 biomedicines-10-02773-f004:**
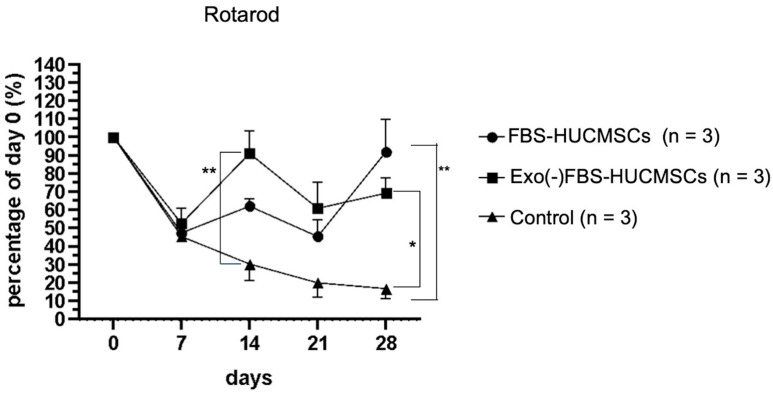
Behavior test (rotarod) of mice treated with exosome-containing (FBS-HUCMSCs) or exosome-depleted FBS-cultured HUCMSCs (Exo(-)FBS-HUCMSCs) or without HUCMSCs treatment (control). The stay time on the rotarod was recorded on days 0, 7, 14, 21, and 28. Day 0 was osteoarthritis induction day. *n* = 3 mice in each group. ** *p* < 0.01, * *p* < 0.05. HUCMSCs were derived from the same donor. FBS: fetal bovine serum.

**Figure 5 biomedicines-10-02773-f005:**
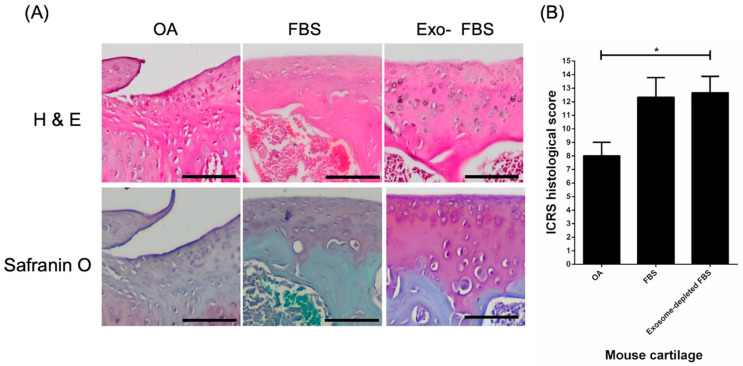
International Cartilage Repair Society (ICRS) score and histology of the mice osteoarthritis (OA) model after 28 days of experiments (*n* = 3 in each group). (**A**) Histology showed by hematoxylin and eosin staining and Safranin O staining of the cartilage in normal saline control, exosome +/− FBS-cultured HUCMSCs-treated knees. Scale bar = 100 μm. (**B**) ICRS histology scores of control and exosome +/− FBS-cultured HUCMSCs-treated knees. The control joints show a significantly lower ICRS histological score than the joints which transplanted 4 × 105 exosome-depleted FBS-cultured HUCMSCs (*n* = 3). * *p* < 0.05. Representative results from 3 experiments. HUCMSCs were derived from the same donor.

**Figure 6 biomedicines-10-02773-f006:**
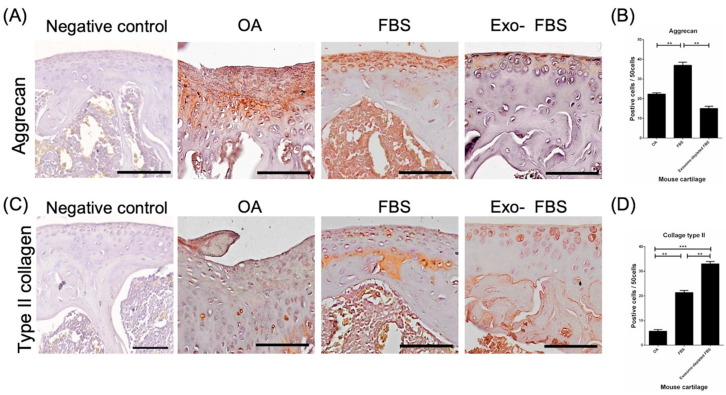
Immunohistochemistry (IHC) of aggrecan and type II collagen in destructed cartilages of mice. (**A**) After 28 days of experiments, the representative image of aggrecan in normal cartilage (IHC negative control), normal saline control, and exosome +/− FBS-cultured HUCMSC-treated knees (*n* = 3 in each group). Scale bar = 100 μm. (**B**) Quantification of aggrecan expression in the three groups. ** *p* < 0.01. (**C**) Representative image of type II collagen in normal cartilage (IHC negative control), normal saline control, and exosome +/− FBS-cultured HUCMSC-treated knees (*n* = 3). Scale bar = 100 μm. (**D**) Quantification of type II collagen expression in the three groups. ** *p* < 0.01, *** *p* < 0.001. Representative results from 3 experiments. HUCMSCs were derived from the same donor.

**Figure 7 biomedicines-10-02773-f007:**
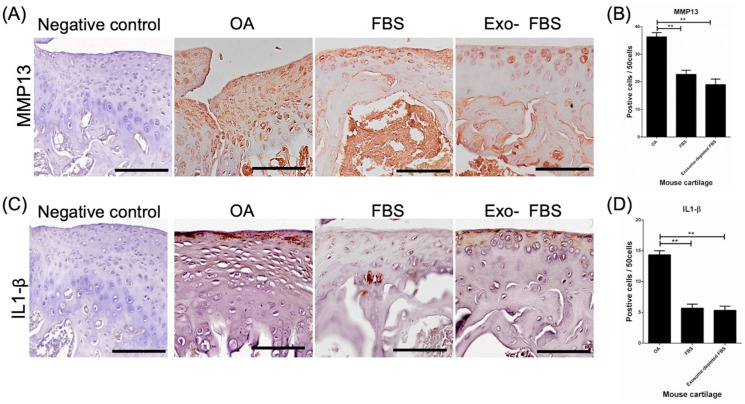
Immunohistochemistry (IHC) of MMP13 and IL-1β in destructed cartilages of mice. (**A**) After 28 days of experiments, a representative image of MMP13 in the normal cartilage (IHC negative control), normal saline control, and exosome +/− FBS-cultured HUCMSC-treated knees (*n* = 3). Scale bar = 100 μm. (**B**) Quantification of MMP13 in the three groups. ** *p* < 0.01. (**C**) Representative image of IL-1β in the normal cartilage (IHC negative control), normal saline control, and exosome +/− FBS-cultured HUCMSC-treated knees (*n* = 3). Scale bar = 100 μm. (**D**) Quantification of IL-1β expression in the three groups. ** *p* < 0.01. Representative results from 3 experiments. HUCMSCs were derived from the same donor.

## Data Availability

All data are presented in the manuscript.

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
