# Peer review of "Chondrogenic Potential of Human Umbilical Cord Mesenchymal Stem Cells Cultured with Exosome-Depleted Fetal Bovine Serum in an Osteoarthritis Mouse Model"

_biomedicines, 2022, doi:10.3390/biomedicines10112773_

Round 1

Reviewer 1 Report

Your manuscript is a interesting proof of the great potential of the umbilical cord in degenrative joint diseases. It looks well written and convincing. The in vitro and in vivo experiments confirm the initial hypothesis. I think that it deserves to be published. Only one thing could be improved. Besides the protocols aproved by the Ethics Commetee for human sampling, the authors could indicate similar approval for animal models.  

Author Response

Your manuscript is a interesting proof of the great potential of the umbilical cord in degenrative joint diseases. It looks well written and convincing. The in vitro and in vivo experiments confirm the initial hypothesis. I think that it deserves to be published. Only one thing could be improved. Besides the protocols aproved by the Ethics Commetee for human sampling, the authors could indicate similar approval for animal models. 

Response: We have added the animal experimental protocol approval statement in the methods section (section 2.11). 

Reviewer 2 Report

In this manuscript, Chang et al investigated the chondrogenic potential of human umbilical cord mesenchymal stem cells cultured with exosome-depleted fetal bovine serum in an osteoarthritis mouse model.

I have specific points to address:

- Introduction section : “However, exosomes in conventional FBS influence the experimental results, including transplantation efficacy.” Please add references

- Introduction section : “Exosome-depleted FBS (Exo(-)FBS) has reduced exosome content compared to conventional FBS”. It seems obvious. Does this mean that it is not possible to fully deplete FBS into exosomes? If so, specify the average percentage of exosome elimination obtained after the exosome depletion procedure.

- Overall, I find that the Introduction section lacks a paragraph explaining why it is important to evaluate the effect of FBS-containing exosomes on the differentiation of HUCMSCs and what benefits can be expected with exosome-depleted FBS compared to undepleted FBS. 

- Result section: “Morphology, proliferation, and surface marker expression levels”. Is it a title? Review formatting

- Section result: “Flow cytometry of HUCMSCs cultured with Exo(-)FBS and FBS showed the same expression pattern”. It would be interesting to add in Figure 1C Dot plot FSC/SSC in order to show that there is no difference in terms of size and granulosity between the 2 conditions. The authors show percentages of positive cells. However, there seem to be important differences in terms of fluorescence intensity of the different markers depending on the conditions. Add MFI. I think it is also important to compare cell viability (Annexin V/PI staining) between the 2 conditions. This should appear in Figure 1.

- Figure 1 and the others: Specify the number of experiments carried out. Fig.1A: Representative results from xx experiments, etc... Please add for each Figure the number of experiments performed. Specify if the experiments were carried out with different sources of HUCMSCs (and add the number of donors) or if they have been performed with cells from the same donor.  Do it for each Figure.

- Figure 2: the authors should quantify cell viability after adipocyte differentiation. Fig.2D: Change ALPL by APAL. Authors should succinctly explain (in Introduction section or in the result section) why they focused on these transcription factors, so that non-expert readers can understand.

-Section “Chondrogenesis of FBS and Exo(-)FBS-cultured HUCMSCs”. Once again, I think it is important to quantify cell viability after chondrogenic differentiation in the 2 cell culture conditions.

- Figure 3A: explain what is 101WJ06

- Section 3.4: “Rotarod behavior after treatment”. In this section, the authors should briefly explain the principle of the Rotarod test and the methodology used as well as the purpose.

- Rework Figure 4: formatting and caption (at least explain what is OA in the caption, or find a more intuitive presentation. FBS in capital letters.

Author Response

In this manuscript, Chang et al investigated the chondrogenic potential of human umbilical cord mesenchymal stem cells cultured with exosome-depleted fetal bovine serum in an osteoarthritis mouse model.

I have specific points to address:

- Introduction section : “However, exosomes in conventional FBS influence the experimental results, including transplantation efficacy.” Please add references

Response: We added a reference [7]. 

- Introduction section : “Exosome-depleted FBS (Exo(-)FBS) has reduced exosome content compared to conventional FBS”. It seems obvious. Does this mean that it is not possible to fully deplete FBS into exosomes? If so, specify the average percentage of exosome elimination obtained after the exosome depletion procedure.

Response: The commercial Exo(-)FBS showed that ≥90% of exosomes were depleted (ThermoFisher).

- Overall, I find that the Introduction section lacks a paragraph explaining why it is important to evaluate the effect of FBS-containing exosomes on the differentiation of HUCMSCs and what benefits can be expected with exosome-depleted FBS compared to undepleted FBS. 

Response: We added a paragraph introducing the importance and benefits of FBS-containing exosomes on the differentiation of HUCMSCs. 

- Result section: “Morphology, proliferation, and surface marker expression levels”. Is it a title? Review formatting

Response: It is a subtitle. We reformatted the title to 3.1.1. 

- Section result: “Flow cytometry of HUCMSCs cultured with Exo(-)FBS and FBS showed the same expression pattern”. It would be interesting to add in Figure 1C Dot plot FSC/SSC in order to show that there is no difference in terms of size and granulosity between the 2 conditions. The authors show percentages of positive cells. However, there seem to be important differences in terms of fluorescence intensity of the different markers depending on the conditions. Add MFI. I think it is also important to compare cell viability (Annexin V/PI staining) between the 2 conditions. This should appear in Figure 1.

Response: We added MFI in Figure 1C, the Dot plot in Figure 1D (CD44), and Annexin V/PI staining in Figure 1E. 

- Figure 1 and the others: Specify the number of experiments carried out. Fig.1A: Representative results from xx experiments, etc... Please add for each Figure the number of experiments performed. Specify if the experiments were carried out with different sources of HUCMSCs (and add the number of donors) or if they have been performed with cells from the same donor.  Do it for each Figure.

Response: We revised them accordingly. 

- Figure 2: the authors should quantify cell viability after adipocyte differentiation. Fig.2D: Change ALPL by APAL. Authors should succinctly explain (in Introduction section or in the result section) why they focused on these transcription factors, so that non-expert readers can understand.

Response: We changed APAL to ALPL. We measured the staining amount to quantify cell viability. The cell viability was decreased after adipogenesis and osteogenesis after cells cultured with Exo(-)FBS (Fig. 2E and F). 

-Section “Chondrogenesis of FBS and Exo(-)FBS-cultured HUCMSCs”. Once again, I think it is important to quantify cell viability after chondrogenic differentiation in the 2 cell culture conditions.

Response: The amount of mRNA was correlated with cell viability. Therefore, we used mRNA amount to represent cell viability after chondrogenesis. Cell viability decreased after being cultured with Exo(-)FBS (Fig. 3E).  

- Figure 3A: explain what is 101WJ06

Response: We added the explanation of 101WJ06. 

- Section 3.4: “Rotarod behavior after treatment”. In this section, the authors should briefly explain the principle of the Rotarod test and the methodology used as well as the purpose.

Response: We added the explanation of the principle of the Rotarod test, its methodology, and its purpose. 

- Rework Figure 4: formatting and caption (at least explain what is OA in the caption, or find a more intuitive presentation. FBS in capital letters.

Response: We revised Figure 4 accordingly.

Round 2

Reviewer 2 Report

The authors answered all my concerns